

# Cold fronts－a potential air quality threat over the Yangtze River Delta, China

Hanqing Kang[1,2,3,4], Bin Zhu[1,2,3,4], Jinhui Gao[5], Yao He[6], Honglei Wang[1,2,3,4], Jifeng Su[7], Chen Pan[1,2,3,4], Tong Zhu[8,9], Bu Yu[10]

[1]Collaborative Innovation Center on Forecast and Evaluation of Meteorological Disaster, Nanjing University of Information Science and Technology, Nanjing, China

[2]Key Laboratory for Aerosol-Cloud-Precipitation of China Meteorological Administration, Nanjing University of Information Science and Technology, Nanjing, China

[3]Key Laboratory of Meteorological Disaster, Ministry of Education (KLME), Nanjing University of Information Science and Technology, Nanjing, China

[4]Joint International Research Laboratory of Climate and Environment Change (ILCEC), Nanjing University of Information Science and Technology, Nanjing, China

[5]Department of Ocean Science and Engineering, Southern University of Science and Technology, Shenzhen, China

[6]Baoji Meteorological Bureau, Baoji, China

[7]The 61 Squad of the 94857 Unit of People's Liberation Army, Wuhu, China

[8]CIRA, Colorado State University, Fort Collins, Colorado, USA

[9]NOAA/NESDIS/STAR/JSCDA, College Park, Maryland, USA

[10]Hangzhou Meteorological Bureau, Hangzhou, China

*Correspondence to*: Bin Zhu (binzhu@nuist.edu.cn)

**Abstract**. Cold frontal passages usually promote quick removal of atmospheric pollutants over North China (e.g. the Beijing–Tianjin–Hebei region). However, in the Yangtze River Delta (YRD), cold fronts pose a potential threat to air quality. In this study, a cold frontal passage and a subsequent stable weather event over YRD during 21–26 January 2015 was investigated with in-situ observations and Weather Research and Forecasting–Community Multiscale Air Quality Modeling System simulations. Observations showed a burst of $PM_{2.5}$ pollution and an obvious southward motion of $PM_{2.5}$ peaks on the afternoon of 21 January, suggesting a strong inflow of highly polluted airmasses to YRD by a cold frontal passage. Model simulations revealed an existing warm and polluted airmass over YRD, which climbed to the free troposphere along the frontal surface as the cold front passed, increasing the $PM_{2.5}$ concentration at high altitudes. Strong north-westerly flow behind the cold front transported particles from the highly polluted North China Plain (NCP) to YRD. As the cold front intruded into the downstream of YRD, high pressure took control over the YRD, which resulted in a synoptic subsidence that brought particles from the free troposphere (1.0–2.0 km) to the surface. After the cold front's passage, weakened winds and a stable atmosphere stayed over the YRD and led to the accumulation of locally emitted $PM_{2.5}$. Tagging of $PM_{2.5}$ by geophysical regions showed that the $PM_{2.5}$



contribution from the YRD itself was 35% and the contribution from the NCP was 29% during the cold frontal passage. However, under the subsequent stable weather conditions, the $PM_{2.5}$ contribution from the YRD increased to 61.5% and the contribution from the NCP decreased to 14.5%. The results of this study indicate that cold fronts are potential bringers of atmospheric pollutants when there are strong air pollutant sources in upstream areas, which may deteriorate air quality in downstream regions.

## 1. Introduction

Fast economic development and urbanization processes in China have led to an increase in air pollution during the past few decades (Han et al., 2016; Chen and Wang, 2015; Cao et al., 2015). Haze, which is formed by fine particulate extinction, has been the most prevalent atmospheric pollution phenomenon over China in recent years (Huang et al., 2014; Wang et al., 2017). The fundamental cause of haze is an increase in particulate matter—especially fine particles—with aerodynamic diameters equal to or less than 2.5 μm ($PM_{2.5}$). Recently, fine particulate matter has caused wide concern owing to its impacts on regional air quality, human health, and climate change.

Densely populated city clusters in China (e.g. the Beijing–Tianjin–Hebei [BTH] region, Yangtze River Delta [YRD], and Pearl River Delta) are associate with heavy particle pollution (Wang et al., 2013b; Liao et al., 2015; Wu, 2007). The two largest city clusters, BTH and YRD, are geographically close to each other. Significant cross-border transport of $PM_{2.5}$ has occurred between BTH and YRD (Li et al., 2013). Cold fronts are important pollutant transport pathways (Liu, 2003; Mari, 2004) that are usually favourable for the quick removal of atmospheric pollutants in BTH (Zhao et al., 2013; Gao et al., 2016). Meanwhile, the YRD is located south of BTH, where cold fronts transport pollutants from BTH to YRD and exacerbate atmospheric pollution. This indicates that the control of emissions in one city cluster is not sufficient to reduce particulate pollution; joint efforts among city clusters are crucial.

The formation mechanisms of $PM_{2.5}$ pollution in China remain highly uncertain owing to complex interactions among pollution sources, meteorology, and atmospheric chemical processes (Guo et al., 2014). Generally, high emission intensity, adverse meteorological conditions, secondary aerosol formation, and the regional transport of particles are main factors contributing to the formation of particulate pollution (Sun et al., 2013; Wang et al., 2014; Wang et al., 2013a; Li et al., 2013). As anthropogenic emissions do not vary much from day to day, particulate pollution episodes are more often associated with adverse meteorological conditions, such as weak surface winds, stable stratified conditions, low mixing layers, and winds from particle source regions that transport large volumes of particles (Tao et al., 2014; Wang et al., 2013a; Li et al., 2017a). Under such weather conditions, substantial amounts of secondary aerosols can be generated and aggravate particulate pollution (Gao et al., 2015; Huang et al., 2014).

Particulate concentrations have been decreasing since 2013 owing to implementation of the Atmospheric Pollution Prevention and Control Action Plan (Wei et al., 2017). However, particle pollution episodes remain frequent, especially in the wintertime. Under the influence of the East Asia winter monsoon, the YRD is dominated by cold air activity in the wintertime. If cold air



activity intensified, cold fronts would intrude into the YRD. In contrast, after a cold front's passage, weakened winds and stable atmosphere remain over the YRD. A regional scale stationary atmosphere is unfavourable for the diffusion of pollutants and leads to haze events, a phenomenon that has been extensively studied over East China (Yang et al., 2015; Wang et al., 2013b; Wang et al., 2013a; Leng et al., 2016). Liu (2003) suggested that frontal lifting to the free troposphere ahead of

southeastward moving cold fronts and transport in the boundary layer behind the cold front are major processes responsible for the export of Asian anthropogenic pollution. Therefore, cold fronts are a potential threat to air quality along its transport pathway.

In this study, we employed the Weather Research and Forecasting (WRF) mesoscale meteorological model and the Community Multiscale Air Quality (CMAQ) modelling system to investigate the sources and formation processes of $PM_{2.5}$ pollution during

a cold frontal passage and subsequent stable weather condition in January 2015. We investigated the formation processes, horizontal distributions, vertical structures, and contributions from source regions to $PM_{2.5}$ over the YRD in both synoptic patterns. Our results highlight the reasons behind high $PM_{2.5}$ episodes and source contributions to $PM_{2.5}$ over the YRD are will be helpful to policy-makers in this region.

**2. Model description and verification**

**2.1 Configuration of weather prediction model**

The numerical model used in this study was the non-hydrostatic, compressible, two-way interactive Advanced Research WRF (version 3.4, Skamarock et al., 2008) coupled with a single-layer urban canopy model (Kusaka et al., 2001; Chen et al. 2004). The simulation domain includes geographical areas (e.g. East China and the Korean Peninsula) with $10 \times 10$ km horizontal resolution and $220 \times 220$ grids, centred at 33.5°N and 118°E (Fig. 1). The vertical grid contains 30 full sigma levels from the

surface to 50 hPa, the lowest 20 levels of which are below 2 km to better resolve processes within the boundary layer. The WRF interior grid-nudging technique was used to improve meteorological fields simulation. An 18-day simulation (from 00:00 UTC 10 January 2015 to 00:00 UTC 28 January 2015) was conducted with initial conditions (ICONs) and boundary conditions (BCONs) from the National Center for Environmental Prediction's 1°-grid-spacing operational Global Forecast System Final Analyses. To represent a more realistic urban land type in the study area, fine resolution (30 s) MODIS 20-category land-use

data were used.

**2.2 Configuration of air quality model**

The CMAQ (version 5.0.2) was applied to simulate gaseous and particulate air pollutants using a 10-km horizontal grid spacing domain that covered East China and Korea (Fig. 1), while the hourly meteorological field was provided by the mesoscale meteorological model WRFv3.4. A period from 10 January 2015 to 28 January 2015 was selected for the simulation, with the

first 9 days the spin-up period to exclude the impacts of uncertainties in ICONs. The ICONs and BCONs for the CMAQ simulation were obtained from the modelling result of the Model for Ozone and Related Chemical Tracers (version 4), an offline global chemical transport model for the troposphere (Emmons et al., 2010). The anthropogenic emissions used in this



study were provided by a mosaic Asian monthly anthropogenic emission inventory, MIX (Li et al., 2017b), with a horizontal resolution of $0.25° \times 0.25°$. Biogenic emissions were generated by the Model for Emissions of Gases and Aerosols from Nature (version 2.1). The CB05 and AERO6 mechanisms were chosen for gas phase chemistry and aerosols, respectively.

The process analysis technique introduced by Gipson (1999) was implemented in the CMAQ modelling system to determine

the contributions of both physical and chemical processes to simulated species. The physical and chemical processes discussed in this study include vertical advection (ZADV), horizontal advection (HADV), vertical diffusion (VDIF), dry deposition (DDEP), cloud processes and aqueous chemistry (CLDS), and aerosol (AERO) processes.

The Integrated Source Apportionment Method (ISAM) has been implemented in CMAQ (Kwok et al., 2013). ISAM tracks contributions from ICONs, BCONs, and user-defined source regions to ambient and deposited gases and aerosol particles.

Currently, ISAM supports two kinds of $PM_{2.5}$ tags: a primary species tag, which tracks the primary emissions of elemental carbon, organic carbon, sulphate, nitrate, ammonium, and other trace elements (e.g. Cl, Na, K, Fe, Ca, Al, Si, Ti, and Mn); and a secondary species tag, which tracks secondarily formed sulphate, nitrate, and ammonium, as well as all gaseous species associated with secondary aerosol species formations (e.g. $SO_2$, $NO_2$, NO, $NO_3$, $HNO_3$, HONO, $N_2O_5$, PAN, and $NH_3$).

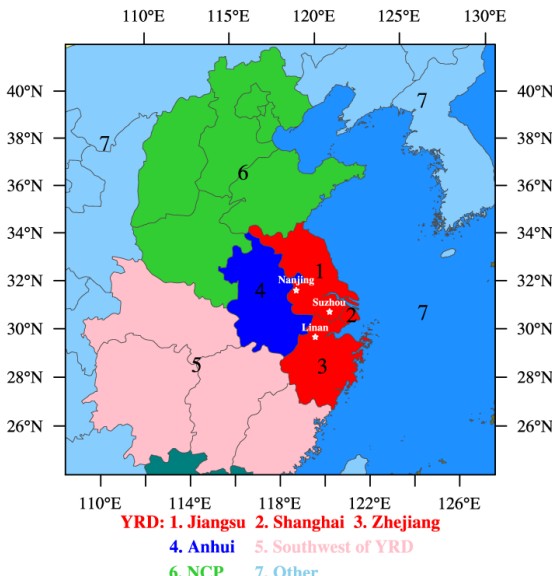

**Figure 1. Modelling domain and source regions. White stars denote the locations of observation sites in Nanjing, Suzhou, and Linan.**

**2.3 Model verification**

The model-simulated surface meteorological parameters and $PM_{2.5}$ concentrations were compared with observations obtained from Nanjing, Suzhou, and Linan (Fig. 1). Figure 2 compares the surface temperature, relative humidity, wind direction, wind speed, and $PM_{2.5}$ concentrations between the simulations and observations from 00:00 LST 19 January 2015 to 00:00 LST 28

January 2015. Simulations from the coupled WRF-CMAQ model appeared to effectively reproduce the variations of meteorological parameters and $PM_{2.5}$ concentrations at the three observation sites.



Some statistical metrics including the correlation coefficient (R), normalized mean bias (NMB), and normalized mean error (NME) were calculated to compare simulated results with observations. The NMB and NME were calculated, respectively, by equations (1) and (2):

$$NMB = \frac{\sum\limits_{i=1}^{N}(M_i - O_i)}{\sum\limits_{i=1}^{N}O_i} \times 100\% \qquad (1)$$

$$NME = \frac{\sum\limits_{i=1}^{N}|M_i - O_i|}{\sum\limits_{i=1}^{N}O_i} \times 100\% \qquad (2)$$

where $M_i$ represents the simulated value, $O_i$ represents the observational data, and $N$ denotes the number of data pairs. Statistical comparisons between the observed and simulated variables are shown in Table 1.

The correlation coefficients for meteorological parameters, except for wind direction and wind speed at Suzhou and Linan, were found to be around 0.90. This discrepancy is likely because the Suzhou station is located in an urban centre, and the Linan station is located on a hill. The $10 \times 10$ km model grid was unable to properly represent the complicated urban canopy at Suzhou and the rolling terrain at Linan. The correlation coefficients for $PM_{2.5}$ concentrations at Nanjing, Suzhou, and Linan were found to be 0.77, 0.68, and 0.74, respectively. This indicates that the time series patterns of $PM_{2.5}$ simulations agree well with observations. The NMB and NME for meteorological parameters were found to be relatively small, except for wind speed and wind direction at Suzhou and Linan. The model systematically underestimated $PM_{2.5}$ concentrations by about 20% for all three stations. This can probably be attributed to the coarse model grid size and lower emission resolution. The NME for $PM_{2.5}$ at all three stations was found to be below 35%, indicating that model performance was acceptable.



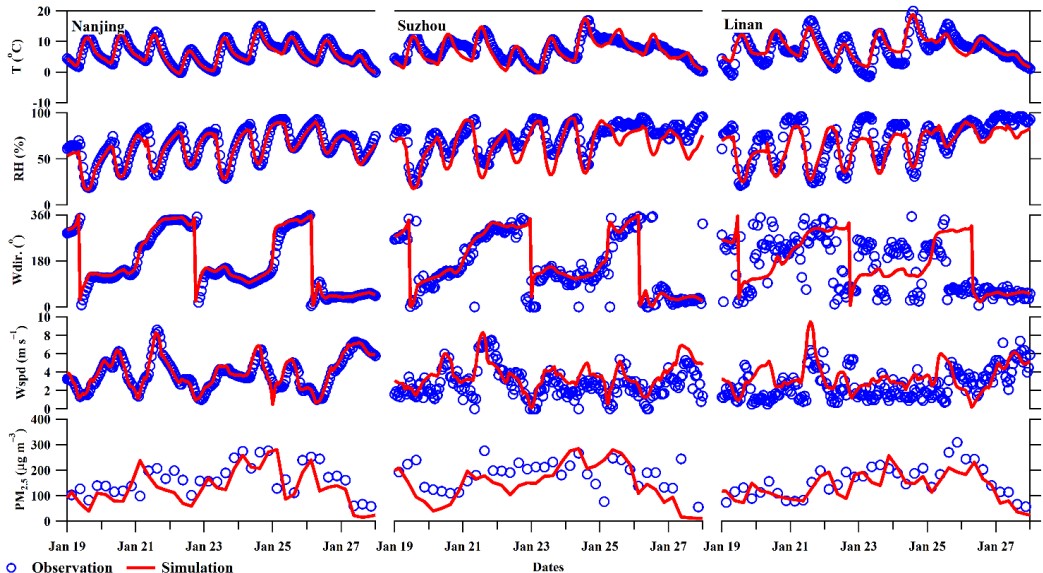

**Figure 2. Observed and simulated 2-m air temperature (T), relative humidity (RH), 10-m wind direction (Wdir), wind speed (Wspd), and surface PM$_{2.5}$ concentrations from 00:00 LST 19 January 2015 to 00:00 LST 28 January 2015 at Nanjing, Suzhou, and Linan.**

## 3. Episode description

The YRD region was suffering from particle pollution in January 2015. The field experiment was carried out from 00:00 LST 19 January 2015 to 00:00 LST 28 January 2015 at Nanjing, Suzhou, and Linan (Fig. 1). The Nanjing observation site is located in a suburban area, the Suzhou station is located in an urban area, and the Linan station represents the regional background site. Observations revealed that 9-day mean PM$_{2.5}$ concentrations reached 100 µg m$^{-3}$ at the Nanjing and Suzhou sites. In some high-pollution episodes, PM$_{2.5}$ concentrations reached as high as 300 µg m$^{-3}$ (Fig. 3).

A short-term burst of PM$_{2.5}$ pollution accompanied by strong northwest winds successively appeared in Nanjing, Suzhou, and Linan between 12:00 LST 21 January 2015 and 04:00 LST 22 January 2015 (Fig. 3). The peaks of PM$_{2.5}$ concentrations reached Nanjing, Suzhou, and Linan at 16:00 LST, 19:00 LST, and 21:00 LST, respectively, with a 5-hour delay from Nanjing to Linan. This process reveals that a strong north-westerly flow brought a polluted airmass across the YRD. Synoptic maps show dense isobars in the head of the cold front, which appeared over the north (upstream) of the YRD at 08:00 LST 21 January 2015 (Fig.

4a). At that moment, a south-westerly wind prevailed in the YRD. Twelve hours later, the cold front moved to the East China Sea (downstream of YRD; Fig. 4b). Meanwhile, the wind direction over the YRD shifted to the north-west, which was favourable for the horizontal transport of air pollutants from the upstream area to the YRD.

After the cold frontal passage, YRD experienced a uniform pressure field for about 3 days (Fig. 4c; Fig. 4d), creating conditions that were unfavourable for the horizontal transport and vertical mixing of atmospheric pollutants (Zhu et al., 2010). Aerosol

particles gradually accumulated over the YRD under this stable atmosphere. In order to exclude the impact of the cold front, this study designated the stable period from 24 to 27 January 2015, when the wind speed was relatively small but PM$_{2.5}$



concentrations were extremely high (Fig. 3), indicating that the pollution likely originated locally. On 27 January 2015, a strong cold front intruded into the YRD accompanied by precipitation, resulting in the significant removal of $PM_{2.5}$.

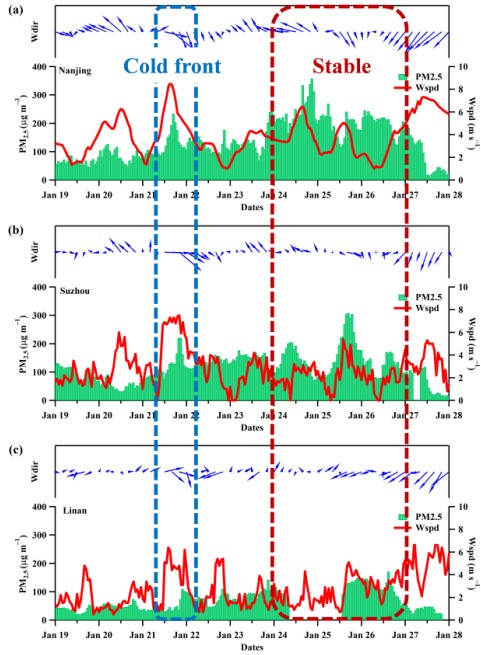

Figure 3. $PM_{2.5}$ concentrations, wind speeds (Wspd), and wind directions (Wdir) at (a) Nanjing, (b) Suzhou, and (c) Linan.

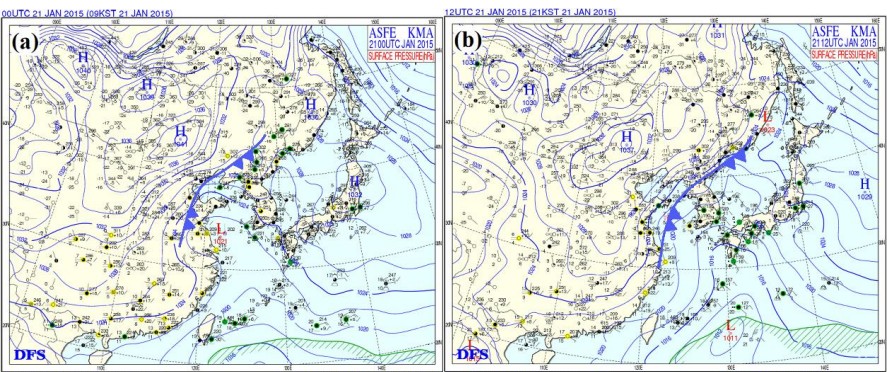



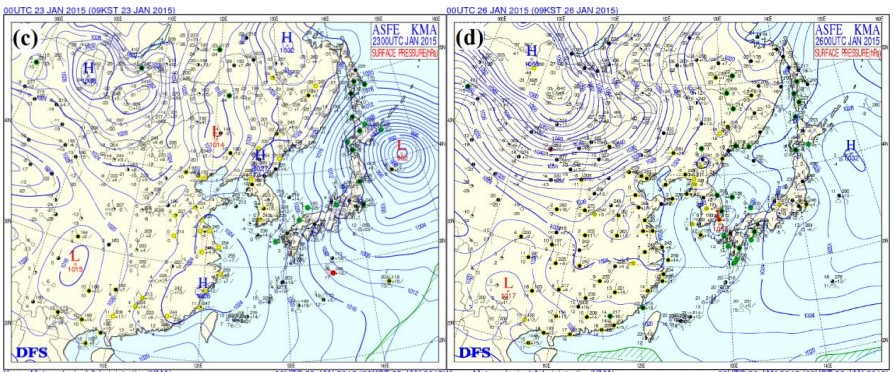

**Figure 4. Surface weather patterns over eastern Asia at (a) 08:00 LST 21 January, (b) 20:00 LST 21 January, (c) 08:00 LST 23 January, and (d) 08:00 LST 26 January.**

**4. Results and discussion**

Observations revealed that the cold front was a carrier of aerosol particles which increased $PM_{2.5}$ concentration over YRD in 21 January. This finding is reproduced by the well-evaluated WRF model. Based on this finding, we considered the formation processes and source contributions of $PM_{2.5}$ pollutions over the YRD during the cold frontal passage and the subsequent stable weather condition.

**4.1 Formation processes of high $PM_{2.5}$ during cold frontal passage**

A strong wind accompanied by high $PM_{2.5}$ concentrations is favourable for the long-range transport of aerosols. Time-averaged $PM_{2.5}$ concentrations and fluxes at the surface and 1.0 km altitude during the cold frontal passage, are shown in Fig. 5. High $PM_{2.5}$ concentrations (> 100 μg m$^{-3}$) and high wind speeds can be observed both at the surface (Fig. 5a) and at 1.0 km (Fig. 5b), resulting in strong $PM_{2.5}$ fluxes from polluted upstream regions to downstream regions. Mean $PM_{2.5}$ fluxes at the surface and at 1.0 km were 619 μg m$^{-2}$ s$^{-1}$ and 1072 μg m$^{-2}$ s$^{-1}$, respectively. $PM_{2.5}$ fluxes were stronger at 1.0 km than at the surface

because the wind speed was higher, while the $PM_{2.5}$ concentrations were comparable to those at surface levels. We can conclude that high altitudes are important aerosol transport pathways during cold frontal passage; they are probably more important than surface transport pathways.





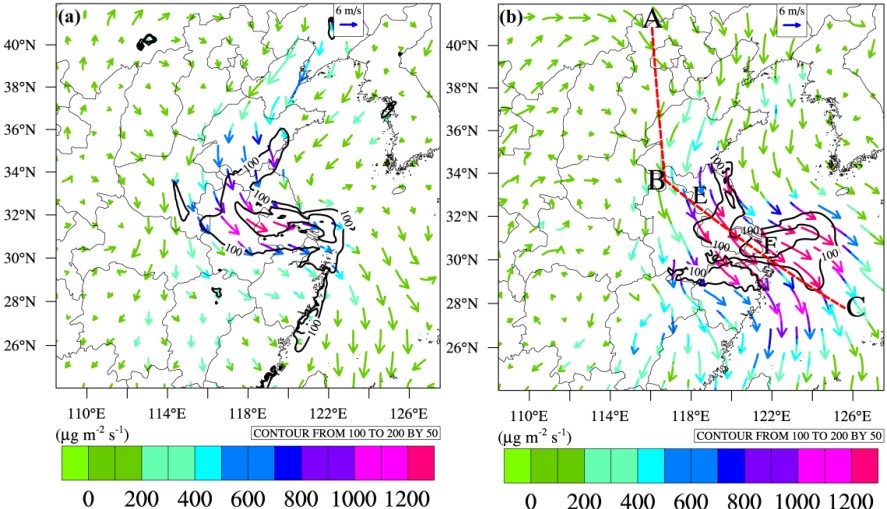

**Figure 5. Mean wind vectors (arrow), PM$_{2.5}$ flux (coloured arrow), and mass concentration (black contour) at (a) the surface and (b) 1.0 km altitude from 12:00 LST 21 January 2015 to 04:00 LST 22 January 2015. The red dashed line (A-B-C) in Fig. 5b denotes the location of the vertical cross section shown in Fig. 6. Points E and F indicate the YRD locations shown in Fig. 6.**

Figure 6 shows a vertical cross section of PM$_{2.5}$ concentration, PM$_{2.5}$ flux, and EPT along the aerosol transport pathway (indicated by the red dashed line in Fig. 5b) during the cold front passage through the YRD from 12:00 LST 21 January to 04:00 LST 22 January. An obvious cold front can be identified over the northern YRD from the EPT contour and wind vectors in Fig. 6a. The EPT contour reveals a stable layer over the YRD with isentropic tilt toward the cold air and parallel to the cold front. Wind vectors show clear downward/upward movements in the north/south of the cold front (red lines in Fig. 6a and 6b).

At noon (12:00 LST) on 21 January, the cold front reached the northern boundary of the YRD accompanied by high PM$_{2.5}$ concentrations (> 100 µg m$^{-3}$) and strong PM$_{2.5}$ fluxes (800−1600 µg m$^{-2}$ s$^{-1}$; Fig. 6a). At the southern end of the cold front, the vertical extent of the high PM$_{2.5}$ concentrations (100 µg m$^{-3}$) reached 2.0 km, significantly higher than the boundary layer height (around 0.6−0.8 km, not shown in Fig. 6a). Therefore, the vertical transport of PM$_{2.5}$ is inferred to be caused by systematic prefrontal upward movements rather than boundary layer turbulent mixing. Surface PM$_{2.5}$ concentrations exceeded

100 µg m$^{-3}$ over the YRD before the cold front's arrival. When the cold front moved into the YRD, it forced the warm and polluted YRD airmass up along the frontal boundary, lifting PM$_{2.5}$ into the upper air (Ding et al., 2009). Liu (2003) suggested that this kind of frontal lifting promotes the transport of pollution to the free troposphere.

In the afternoon (16:00 LST) of 21 January, the cold front intruded into the YRD (Fig. 6b). A deep neutral stratified condition appeared over the YRD because of the strong wind. The high PM$_{2.5}$ concentration zone moved south alongside the cold front.

Aerosols from NCP were transported to the YRD by strong north-westerly flow; hence, increased aerosol concentrations and fluxes over the YRD.

At the end of the cold frontal period (Fig. 6c; Fig. 6d), when the cold front moved to downstream of the YRD, the YRD was



under a high-pressure system that resulted in divergence (e.g. the vertical PM$_{2.5}$ flux at 1.0 km was about –0.9 µg m$^{-2}$ s$^{-1}$, and at 0.5 km was about –0.5 µg m$^{-2}$ s$^{-1}$). Synoptic subsidence behind the cold front would suppress the upward transport of PM$_{2.5}$, increasing the surface particle concentration (Mari, 2004). Downward motions would bring particles from the free troposphere (1.0–2.0 km) down to the surface and trap pollutants in the boundary layer. Additionally, an extremely strong southward PM$_{2.5}$

5   flux (> 1800 µg m$^{-2}$ s$^{-1}$) can be identified over the YRD, indicating the transport pathway of PM$_{2.5}$ (Fig. 6c). Up until the next morning (04:00 LST 22 January), high PM$_{2.5}$ concentrations were primarily restricted to below 1.0 km over the YRD (Fig. 6d). The high concentration of PM$_{2.5}$ that appeared over both the YRD and its downstream regions was probably due to the mixing of locally emitted particles with those brought by the cold front from the NCP.

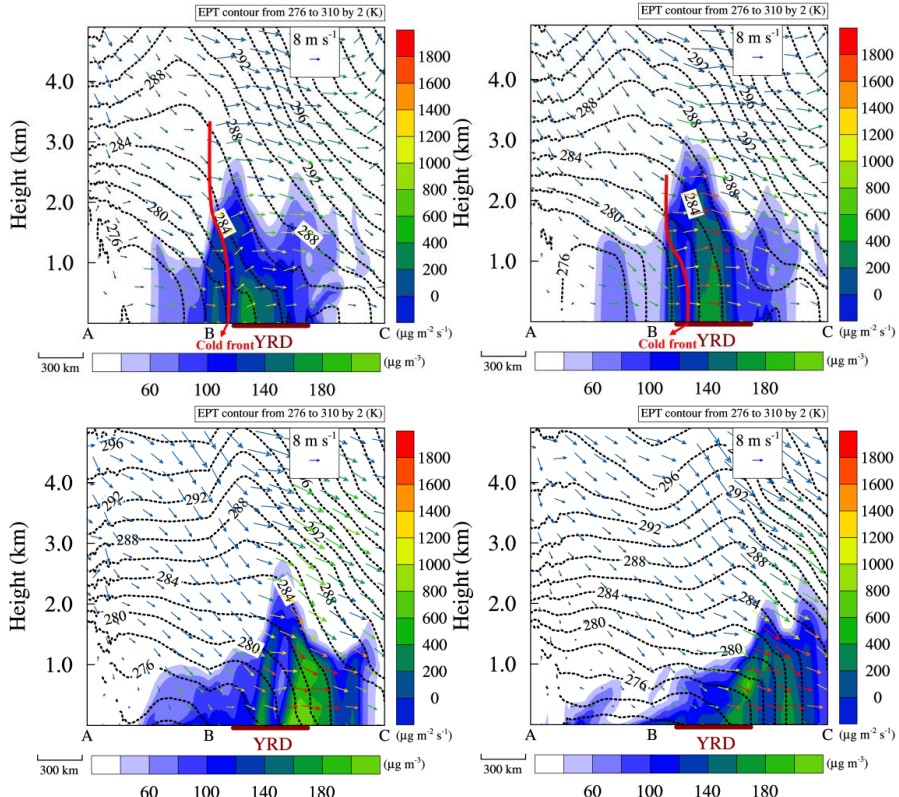

**Figure 6. Vertical cross sections of PM$_{2.5}$ concentration (colour-filled contour), EPT (dashed black lines), in-plane wind vectors (arrow) where the vertical speed is multiplied by 100, and PM$_{2.5}$ flux (arrow colour), at (a) 12:00 LST 21 January, (b) 16:00 LST 21 January, (c) 22:00 LST 21 January, and (d) 04:00 LST 22 January. The thicker red lines in (a) and (b) denote the locations of the cold front.**

15  A process analysis technique was introduced to evaluate the effects of physical and chemical processes on aerosol vertical distributions over the YRD. Figure 7 shows the profiles of the averaged PM$_{2.5}$ concentrations and the contributions of VDIF, AERO, ZADV, and HADV processes to PM$_{2.5}$ concentrations over the YRD during the cold frontal passage. At the beginning of the cold front period (12:00 LST to 16:00 LST 21 January), the contributions of vertical advection processes to PM$_{2.5}$



concentrations were negative (decreased aerosol concentrations) below 1.0 km, but positive (increased aerosol concentrations) between 1.0 km and 2.5 km (Fig. 7a). This supports the previous conclusion that vertical motions lifted particles from the boundary layer to the free troposphere during the cold frontal passage.

The horizontal advection process increased $PM_{2.5}$ concentrations below 1.0 km but decreased $PM_{2.5}$ concentrations above 1.0

km. Through the horizontal advection process, the cold airmass brought aerosols from the NCP to the YRD, increasing the surface aerosol concentration over the YRD. The negative contribution of horizontal advection above 1.0 km was probably due to aerosol concentrations being increased by strong prefrontal lifting that transported aerosols from the surface to the free troposphere, thus strengthening the outflow of free-tropospheric aerosols from the YRD. The vertical diffusion process has a relatively small effect on aerosol vertical distributions except for in the first layer, where most of the emissions exist. Vertical

aerosol concentrations were slightly increased through secondary aerosol formation.

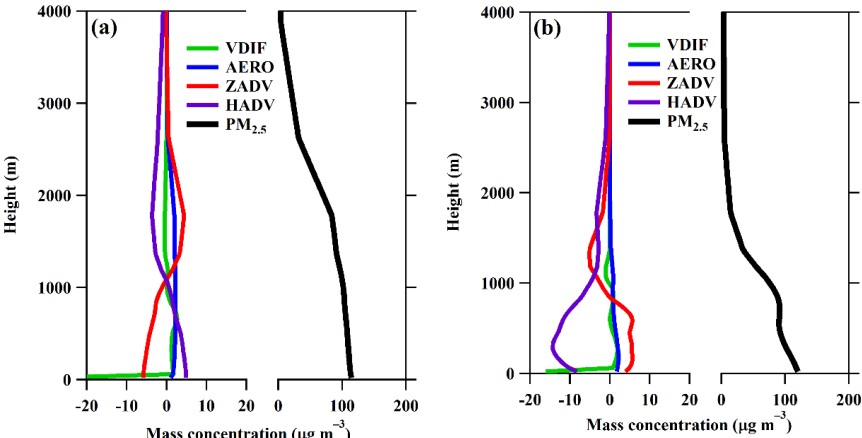

**Figure 7. Vertical profiles of $PM_{2.5}$ concentrations and the contributions of physical/chemical processes to $PM_{2.5}$ concentrations (a) at the beginning and (b) at the end of the cold frontal period.**

The profiles of averaged $PM_{2.5}$ concentrations and the contributions of physical and chemical processes to $PM_{2.5}$ concentrations

over the YRD at the end of the cold frontal period (22:00 LST 21 to 04:00 LST 22 January) are shown in Fig. 7b. The vertical advection process made positive contributions to $PM_{2.5}$ concentrations in the lower atmosphere but negative contributions in the upper atmosphere—the opposite of the result obtained at the beginning of cold frontal passage. This result supports our previous conclusion that divergence after the cold front (Fig. 6c; Fig. 6d) transports particles from the free troposphere to the boundary layer. The contributions of the horizontal advection process were negative between the surface and the free

troposphere, implying a net horizontal outflow of aerosols from the YRD. At this time, the upstream of the YRD was cleaner than the YRD itself.

**4.2 Formation processes of high $PM_{2.5}$ concentrations under stable weather**

After the cold frontal passage, aerosol particles started to accumulate under stable atmosphere that resulted in high $PM_{2.5}$





concentrations in the near-surface layer over east China (Fig. 8a). In the centre of the YRD (including south of Jiangsu and north of Zhejiang), the mean $PM_{2.5}$ concentration was more than 200 $\mu g\ m^{-3}$ higher than that of cold front period, but the $PM_{2.5}$ concentrations at 1.0 km (Fig. 8b) were significantly lower. The $PM_{2.5}$ fluxes in the stable atmosphere were lower than those in cold frontal passage at both the surface and 1.0 km, reflecting the lower mean wind velocity under stable weather. This

indicates that atmospheric conditions were not favourable for the horizontal transport of $PM_{2.5}$.

The averaged $PM_{2.5}$ profile over the YRD shows significant vertical gradients under the stable weather (Fig. 9). The process analysis showed that the vertical advection process transported $PM_{2.5}$ from the surface to the upper air. However, this vertical transport only reached 1.0 km altitude—much lower than it did during the cold frontal passage (~ 2.0 km). Horizontal advection shows a small negative contribution to $PM_{2.5}$ concentrations over the YRD from the surface to 1.0 km. This indicates that there

was a weak outflow of $PM_{2.5}$ from the YRD to its surroundings, because the YRD is an important aerosol source region. Vertical diffusion mixing $PM_{2.5}$ between the surface and the upper air, but its contribution to $PM_{2.5}$ was relatively small. Secondary aerosol formation slightly increased the aerosol concentration from the surface to 1.0 km.

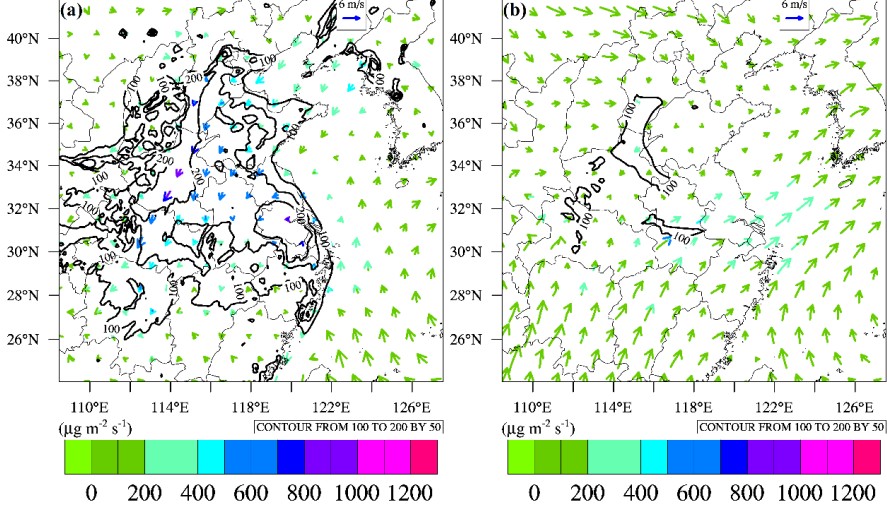

**Figure 8. Averaged $PM_{2.5}$ flux (coloured arrows) and mass concentrations (black contour) at (a) the surface and (b) 1.0 km altitude**

**from 00:00 LST 24 January 2015 to 00:00 LST 27 January 2015.**



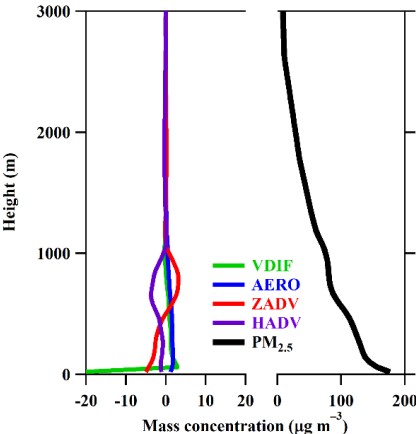

**Figure 9. Vertical profiles of PM$_{2.5}$ concentrations and the contributions of physical/chemical processes to PM$_{2.5}$ concentrations during the period of stable weather.**

**4.3 Contributions of PM$_{2.5}$ from source regions to YRD**

Anthropogenic emissions are the fundamental source of PM$_{2.5}$. Knowing the source regions of aerosols and their contributions to the YRD is critical in controlling PM$_{2.5}$ pollution. Our results revealed a significant transport of aerosol particles from the NCP to the YRD during the cold frontal passage and a remarkable local PM$_{2.5}$ contribution during stable weather conditions. Based on these results, we derived the contributions of PM$_{2.5}$ from source regions to the YRD using ISAM, which was incorporated in the CMAQ model.

Mass contributions from each of the geographical source regions, BCONs, and ICONs to PM$_{2.5}$ concentrations over the YRD from 19 to 28 January 2015 are shown in Fig. 10. The YRD is a quickly developing and densely populated region where anthropogenic activities such as industrial production, vehicle usage, power plant operation, and residential happenings release huge volumes of atmospheric pollutants. Therefore, in the YRD, the most significant source of PM$_{2.5}$ is local emissions (Fig. 10). The NCP is another heavily polluted region in east China (Cao et al., 2015, Chen and Wang, 2015, Li et al., 2017), located

adjacent to YRD to the south. In 21 January, a cold front brought polluted airmass from the NCP to the YRD resulting in a high contribution of PM$_{2.5}$. PM$_{2.5}$ from regions outside of the modelling domain (BCONs) also impacted on the PM$_{2.5}$ concentration over the YRD through long-range transport. However, contributions from other source regions were relatively small.

   Mean contributions from each source region from 19 to 28 January 2015 are shown in Fig. 11a. Local contributions (from

the YRD itself) accounted for 56.5% of the PM$_{2.5}$ concentration, in which Jiangsu, Shanghai, and Zhejiang accounted for 32.5%, 3.5%, and 20.5%, respectively. PM$_{2.5}$ from the NCP and BCONs contributed 18.5% and 10.5%, respectively. The YRD, NCP, and BCONs contributed 85.5% in total.

During the cold frontal passage, a strong northwest wind prevailed over the YRD; locally originated aerosols only accounted





for 35% of the PM$_{2.5}$ contribution, with Jiangsu, Shanghai, and Zhejiang accounting for 22%, 2%, and 11%, respectively (Fig. 11b). These contributions are much lower than those in the total average because the strong wind in the cold front period was unfavourable to the accumulation of locally emitted pollutants. PM$_{2.5}$ from the NCP contributed 29% to the PM$_{2.5}$ concentrations over YRD, a significantly higher amount than in the average contribution. Contributions from other

geographical source regions were also increased during the cold frontal passage because of the long-range transport of aerosol. In general, the cold front decreased local contributions in the YRD, but increased long-range transport contributions from the NCP region.

Under stable weather conditions, local contributions (61.5%) were increased, especially for Zhejiang province (Fig. 11c). Lower wind speeds during the stable period were unfavourable to the transport of pollutants, resulting in high PM$_{2.5}$

concentrations and significant local contributions. NCP contributed 14.5% to PM$_{2.5}$ concentrations over the YRD, accounting for only half that in the cold front period. In general, PM$_{2.5}$ contributions in the stable period were similar to those in the total average, but with higher local contributions and lower NCP contributions.

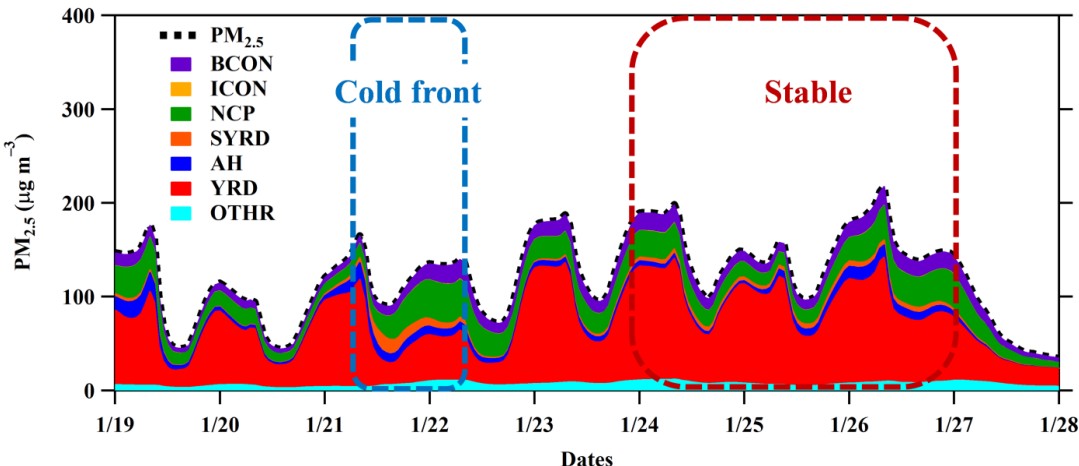

**Figure 10. Time series of PM$_{2.5}$ concentrations and contributions of source regions to the PM$_{2.5}$ concentrations over the Yangtze**

**River Delta (YRD) from 19 to 28 January 2015. Southwest of YRD = SYRD, Anhui = AH, and Other = OTHR.**

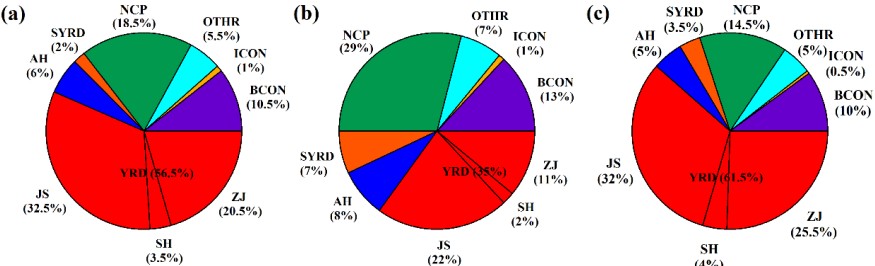

**Figure 11. Contribution rate of each source region to PM$_{2.5}$ over the Yangtze River Delta (YRD) during (a) the whole simulation period, (b) the cold frontal passage, and (c) under the stable weather conditions. Jiangsu = JS, Shanghai = SH, and Zhejiang = ZJ.**

**5. Conclusions**

Cold fronts are important PM$_{2.5}$ transport pathways in wintertime, removing aerosol particles as soon as they reach the BTH. However, in the YRD, cold fronts remove local aerosol particles and can also introduce upstream pollutants. Understanding the processes of PM$_{2.5}$ transport during cold frontal passage is of great significance for the understanding of haze formation

mechanisms over the YRD in wintertime. In this study, the coupled WRF-CMAQ model was employed to investigate the processes and mechanisms of PM$_{2.5}$ pollution over the YRD under a cold frontal intrusion period and a subsequent stable weather conditions in January 2015.

Three sites' observations show that high PM$_{2.5}$ concentrations and strong north-westerly winds appeared simultaneously as the locations of the peak PM$_{2.5}$ concentration moved from north to south, indicating that the cold front transported aerosol particles

across the YRD. At the beginning of the cold frontal passage, when the cold front first reached the YRD, it forced the warm and polluted YRD airmass to move up along the frontal boundary, lifting PM$_{2.5}$ into the free troposphere. As the cold front intruded deep into the YRD, aerosols from upstream areas (NCP) were transported to the YRD by strong north-westerly flow. At the end of the cold front period, when the cold front had moved to the downstream area (East China Sea), the YRD fell under high pressure, resulting in divergence over the region. The synoptic subsidence motions brought particles from the free

troposphere (1.0–2.0 km altitude) to the surface and trapped PM$_{2.5}$ in the boundary layer. The atmospheric stratification became stable after the cold front from 24 to 27 January 2015. Aerosol particles over the YRD then began to reaccumulate until the next cold front.

The contributions of PM$_{2.5}$ from each of the defined source regions were calculated by ISAM. For the entire 9-day simulation (19–28 January), PM$_{2.5}$ contributions from the local area (YRD), NCP, and BCONs accounted for 56.5%, 18.5%, and 10.5%,

respectively. During the cold front passage (12:00 LST 21 January to 04:00 LST 22 January), local PM$_{2.5}$ contributions decreased to 35%, while contributions from the NCP increased to 29%. During the stable weather conditions (00:00 LST 24 January to 00:00 LST 27 January), local PM$_{2.5}$ contributions increased to 61.5% while NCP contributions decreased to 14.5%. This result indicates that cold fronts are a potential carrier of atmospheric pollutants, and may contribute significantly to PM$_{2.5}$ concentrations in downstream areas.

**Acknowledgements**

This work was supported by the National Natural Science Foundation of China (Grant No. 91544229, 41605091 and 41605096), the National Key Research and Development Program (Grant No. 2016YFA0602003), The Startup Foundation for Introducing Talent of NUIST (Grant No. 2243141501035), the Hangzhou Scientific Research Project in Agriculture and Social Development (Grant No. 20170533B16), and the Open fund by the Key Laboratory for Aerosol-Cloud-Precipitation of CMA-

NUIST (Grant No. KDW1701). We acknowledge the free use of MIX emission from Tsinghua University.

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

Tables

**Table 1. Statistical comparisons between the observed and simulated meteorological parameters and PM$_{2.5}$ concentrations at Nanjing, Suzhou, and Linan[a].**

|  | Nanjing | | | Suzhou | | | Linan | | |
| --- | --- | --- | --- | --- | --- | --- | --- | --- | --- |
|  | R | NMB | NME | R | NMB | NME | R | NMB | NME |
| T | 0.97 | -4.9% | 15% | 0.90 | -4.3% | 23.6% | 0.90 | 12.9% | 30.1% |
| RH | 0.94 | -5.1% | 7.7% | 0.84 | -9.7% | 13.4% | 0.85 | -9.7% | 14.8% |
| Wdir | 0.89 | 5.7% | 12.2% | 0.80 | 7.3% | 22.6% | 0.39 | 14.6% | 54.3% |
| Wspd | 0.94 | 2.0% | 11.2% | 0.68 | 37.2% | 45.4% | 0.37 | 37.4% | 61.7% |
| PM$_{2.5}$ | 0.77 | -23.1% | 32.0% | 0.68 | -21.9% | 34.8% | 0.74 | -17.9% | 27.8% |

a.    R = correlation coefficient, NMB = normalized mean bias, NME = normalized mean error, T = air temperature, RH = relative humidity, Wdir = wind direction, Wspd = wind speed, PM$_{2.5}$ = PM$_{2.5}$ concentration.