# Peer review of "Potential impacts of cold frontal passage on air quality over the Yangtze River Delta, China"

_Atmospheric Chemistry and Physics, 2018_

## Referee Comment (RC1) · Anonymous Referee #1 · 8 Nov 2018

This manuscript presents a very interesting story combing synoptic weather and transport of air pollutants. Both process analysis and source apportionment techniques can confirm the results, which are scientifically solid. I only have minor concerns on the presentation of results. Specific comments are listed below:

1. Page 2, line 7: references should be added here.

2. Page 2, line 32: "Liu (2003)..." this sentence is very confusing. Please make it more clear.

3. Page 8, line 20: EPT should be defined.

4. Page 9, line 2: please indicate how is cold front (red line) is diagnosed and plotted?

5. Fig 6: the color scale is too large. I would suggest using 0-500

---

## Referee Comment (RC2) · Anonymous Referee #2 · 9 Nov 2018

This paper tries to reveal impacts of synoptic process on local/regional air quality. The authors focus on cold fronts and their related weather process. This kind of large scale weather system can of course influence local air quality, during its different stage of passing the concerned site. The discussion in this paper is of scientific meaning.

My major concern is on the title, since a cold front itself cannot be a threat on air quality. Usually we consider the passage of a cold front as a cleaner to local air pollutants, since stronger winds may accompany to the cold front. Therefore the title is misleading.

Because of the title and related conceptual confusion, many phenomena and processes are not described/interpreted properly in this paper. It happens also for some concluding sentences.

[Figure]

In addition, this is only a case study for 18 days of weather processes.

Other points:

1)Page 1 line 29: "The results of this study indicate that cold fronts are potential bringers of atmospheric pollutants. . .", not exactly real.

2)Page 2 line 11: "Cold fronts are important pollutant transport pathways", what does 'pathway' mean here?

3)Page 3 line 1: " Therefore, cold fronts are a potential threat to air quality along its transport pathway", this is never a logic conclusion to previous sentences.

4)Page 8 line 3: " Observations revealed that the cold front was a carrier of aerosol particles which increased PM2.5 concentration over YRD in 21 January. This finding. . .", The cold front carries the air pollutants? Or just the fact is that the air pollutants accompanies the cold front?

5)Page 10 line 14: "Through the horizontal advection process, the cold airmass brought aerosols from the NCP to the YRD", what is the evidence?

6)Page 14 line 8: "Cold fronts are important PM2.5 transport pathways", Cold front is. . . transport pathway? What do you mean?

---

## Author Comment (AC1) · 30 Dec 2018

This manuscript presents a very interesting story combing synoptic weather and transport of air pollutants. Both process analysis and source apportionment techniques can confirm the results, which are scientifically solid. I only have minor concerns on the presentation of results. Specific comments are listed below:

1. Page 2, line 7: references should be added here.

Response: Thanks for your suggestion. We added 3 references here. Please refer to page 2, line 12 in the paper.

2. Page 2, line 32: "Liu (2003). . ." this sentence is very confusing. Please make it more clear.

Response: Thanks for your suggestion. We replaced this sentence by the original sentence of the cited paper. It's more clear now. Please refer to page 3, line 5 in the paper.

3. Page 8, line 20: EPT should be defined.

Response: Thanks for your suggestion. EPT is equivalent potential temperature. We added the definition in page 9 line 5.

4. Page 9, line 2: please indicate how is cold front (red line) is diagnosed and plotted?

Response: The cold front is diagnosed from the densely EPT contour near surface and the significant vertical wind shear. The cold front (red line) is manually plotted.

5. Fig 6: the color scale is too large. I would suggest using 0-500.

Response: Thanks for your suggestion. Actually, the color scale is only large in Fig. 6a. In Fig. 6c and Fig. 6d, the maximum PM2.5 fluxes are larger than 2000 ug m-2 s-1, if we use 0-500 color scale, we can not distinguish the differences of PM2.5 fluxes between surface and up level.

---

## Author Comment (AC2) · 30 Dec 2018

Anonymous Referee #2 Received and published: 9 November 2018

This paper tries to reveal impacts of synoptic process on local/regional air quality. The authors focus on cold fronts and their related weather process. This kind of large scale weather system can of course influence local air quality, during its different stage of passing the concerned site. The discussion in this paper is of scientific meaning. My major concern is on the title, since a cold front itself cannot be a threat on air quality. Usually we consider the passage of a cold front as a cleaner to local air pollutants, since stronger winds may accompany to the cold front. Therefore the title is misleading. Because of the title and related conceptual confusion, many phenomena and processes are not described/interpreted properly in this paper. It happens also for some concluding sentences. In addition, this is only a case study for 18 days of weather processes.

Response: Thanks for your comments. Yes, you are right, local air pollutants can be cleaned by cold front because of the stronger winds. But cold front can also bring air pollutants from upstream areas. Lin et al. (2007) suggested that long-range transport of Asian dust and upstream air pollutants by cold fronts are important environmental issues of Taiwan during the winter monsoon season. Liu (2003); Ding et al. (2009) also pointed out that frontal activity plays important roles in the long-range transport of air pollutants.

In this study, we found a series of air pollution episodes accompanied by cold frontal passage over YRD from December 2014 to February 2015 (Figure below, not shown in the manuscript, but we added descriptions to the manuscript in page 6 line 18-21.). At least 13 cold frontal cases (thicker black numbers in the figure) were found accompanied by the long-range transport of PM2.5 from NCP to YRD. During these cold frontal cases, air pollution over YRD always occurred 1-2 days later than that over NCP and PM2.5 concentrations elevated drastically along with the strong northerly winds. It is very clear that the rapid increase of PM2.5 concentrations over YRD in a short period in these cases (case 6, 7, 8, 9 and 11 are more obvious than others; in this study, we chose case 8 and the following local accumulation air pollution episode for further discussion) are attributed to cold fronts' transport rather than locally accumulation. If there are no cold frontal passage in these cases, PM2.5 concentrations over YRD would be much lower. Therefore, cold fronts are potential air quality threats over YRD. According to above discussions and conclusions, we think that the title of this paper is not misleading and the concept is clear. Note that cold front generally deteriorates air quality over YRD in a short time period, but it will finally clean the atmosphere. This study provided a new insight into the understanding of air pollution formation mechanisms over YRD.
Besides, we thoroughly revised the full text, make sure all phenomena and processes are properly described/interpreted.

Other points:

1)Page 1 line 29: "The results of this study indicate that cold fronts are potential bringers of atmospheric pollutants. . .", not exactly real.

Response: Thanks for your comment. From the result of this study and the picture below, we can conclude that cold fronts are potential bringers of atmospheric pollutants over YRD. But in other places, this conclusion may not be true.

2)Page 2 line 11: "Cold fronts are important pollutant transport pathways", what does 'pathway' mean here?

Response: We changed this sentence to "Cold fronts are important ways of pollutant transport." in page 2 line 16.

3)Page 3 line 1: "Therefore, cold fronts are a potential threat to air quality along its transport pathway", this is never a logic conclusion to previous sentences.

Response: Thanks for your comment. We changed this sentence to "Therefore, cold fronts may have significant impact on air quality along its transport pathway." in page 3 line 8.

4)Page 8 line 3: "Observations revealed that the cold front was a carrier of aerosol particles which increased PM2.5 concentration over YRD in 21 January. This finding. . .", The cold front carries the air pollutants? Or just the fact is that the air pollutants accompanies the cold front?

Response: In this cold frontal case, air pollution was formed in NCP, then transported to YRD by cold frontal intrusion. We changed this sentence to "Observations revealed that the cold front pushed polluted airmasses over NCP to YRD, which increased PM2.5 concentration over YRD in 21 January." in page 8 line 5.
5)Page 10 line 14: "Through the horizontal advection process, the cold airmass brought aerosols from the NCP to the YRD", what is the evidence?

Response: Process analysis technique can provide contributions of each physical/chemical process to PM2.5 concentrations over YRD. Contributions from horizontal advection process are positive near surface (Fig. 7a) that means PM2.5 horizontal inflows are stronger than outflows. The horizontal inflow of PM2.5 mainly comes from NCP, which can be found from the PM2.5 fluxes in Fig. 6.

6)Page 14 line 8: "Cold fronts are important PM2.5 transport pathways", Cold front is. . . transport pathway? What do you mean?

Response: We change this sentence to "Cold fronts are important ways of PM2.5 transport" in page 15 line 2.

Ding, A., Wang, T., Xue, L., Gao, J., Stohl, A., Lei, H., Jin, D., Ren, Y., Wang, X., Wei, X., Qi, Y., Liu, J., and Zhang, X.: Transport of north China air pollution by midlatitude cyclones: Case study of aircraft measurements in summer 2007, J. Geophys. Res., 114, doi:10.1029/2008jd011023, 2009. Lin, C.-Y., Wang, Z., Chen, W.-N., Chang, S.-Y., Chou, C. C. K., Sugimoto, N., and Zhao, X.: Long-range transport of Asian dust and air pollutants to Taiwan: observed evidence and model simulation, Atmospheric Chemistry and Physics, 7, 423-434, doi:10.5194/acp-7-423-2007, 2007. Liu, H.: Transport pathways for Asian pollution outflow over the Pacific: Interannual and seasonal variations, J. Geophys. Res., 108, doi:10.1029/2002jd003102, 2003.

Fig. 1. Observed surface PM2.5 concentrations (color) and wind vectors (only the wind speeds greater than 3 m/s are shown) at 14 sites from 1 December 2014 to 28 February 2015. Labels on left axis are latitudes of the 14 observation sites. Red numbers are cold frontal episodes that transport PM2.5 from NCP to YRD.

---

## Author Response (AR2)

**Comments and responses**

This version is much better than previous one.

But following sentences are still vague:

1) page 1, line 21: "However, in the Yangtze River Delta (YRD), cold fronts pose a potential threat to air quality". Cold fronts threat air quality?

**Response:** Thanks for your suggestion. We changed this sentence to "However, in the Yangtze River Delta (YRD), cold fronts may bring air pollutants from the polluted North China Plain (NCP), thereby deteriorating the air quality in the YRD.". Please refer to page1, line 21.

2) page 2, line 15: "Cold fronts are important ways of pollutant transport". Cold fronts are …transport ways?

**Response:** Thanks for your suggestion. We changed this sentence to "Cold fronts promote the long-range transport of dust and anthropogenic air pollutants". Please refer to page 2, line 16.

3) page 6, line 20: "That means cold front frequently exacerbates air pollution over YRD in wintertime, however, we did not notice it before." Cold front exacerbates air pollution?

**Response:** Thanks for your suggestion. We changed this sentence to "That means long-range transport of air pollutants by cold frontal passage may be an important cause of air pollution over YRD in wintertime, however, we did not notice it before.". Please refer to page 6, line 20.

4) page 15, line 2-3: "Cold fronts are important ways of PM2.5 transport in wintertime, removing aerosol particles as soon as they reach the BTH". "cold fronts remove local aerosol particles and can also introduce upstream pollutants", Cold fronts are …transport ways again? And other roles?

**Response:** Thanks for your suggestion. This sentence was changed to "Cold fronts are favourable to the outflow of $PM_{2.5}$ in the BTH, decreasing aerosol concentrations as soon as they reach the area.". Please refer to page 15, line 2. Other roles of the cold front please refer to the response to comment 6.

5) page 15, line 23: "This result indicates that cold fronts are a potential threat on air quality over YRD". Cold fronts are a potential threat on air quality again?

**Response:** Thanks for your suggestion. This sentence was changed to "This result indicates that cold fronts intensify the long-range transport of air pollutants in the NCP to the YRD.". Please refer to page 15, line 23.

6) I do not satisfy the authors response about the role of cold front, which has complex structure. The explanation and conclusion in this paper about the cold front and its role playing to local air quality is over simplified. My opinion is, as a synoptic phenomenon, cold front itself cannot be a threat to air quality, if there is no emission of air pollutants. In all literatures the authors mentioned, the phase of "cold front activity" or "cold front passage" is used, rather than cold front itself. Actually, cold front play multiple roles in air quality, since it relates to strong wind, enhancement of turbulence, boundary layer deformation, air lifting and subsidence, etc. Therefore, strong wind and air lifting may be favorite to long-range transport of air pollutants; enhanced turbulence may increase dust emission from the surface (e.g., in case of sand storm); boundary layer depression may help accumulation of air pollutants, and so on. In the case of this paper, cold front passage and its related processes (transport), do have impacts on the air quality of YRD, as the results have shown.

So, I suggest the paper title changes to "Potential impacts of cold front passage on air quality of YRD, …".

I also suggest to clearly indicate what part of the cold front plays respective roles. For example, the air mass behind the cold front? The airflow ahead of the front? Or just the surface frontal zone? And furthermore, how the frontal passage and frontal structure impact the local air quality?

**Response:** Thanks for you detailed comments and suggestions. Yes, just like you said, cold front has complicate impacts on air quality. In this study, we focused on the "transport" function of cold front. Vertical diffusion profile in process

analysis shows that its contribution to $PM_{2.5}$ is small (Fig. 7), indicating that turbulence has little effect on vertical distributions of $PM_{2.5}$ in this cold front episode. Boundary-layer deformation, air lifting and subsidence show different characteristics in different parts of the cold front. For example, warm and polluted airmass ahead of the frontal zone moved up along the frontal boundary, lifting $PM_{2.5}$ into the free troposphere, resulted in a smaller $PM_{2.5}$ vertical gradient (Fig. 7a); strong frontal north-westerly airflow transported $PM_{2.5}$ from NCP to YRD; synoptic subsidence motions behind the frontal zone trapped $PM_{2.5}$ in the surface, resulted in a larger $PM_{2.5}$ vertical gradient (Fig. 7b). With the passage of cold front, the YRD has experienced all these processes. Follow your suggestions, we used different "frontal zone" instead of "cold front passage" or "cold front activity". Please refer to page 1, line 27, 28; page 9, line 17; page 11, line 5, 21; page 15, line 11, 13.

Follow your suggestion, we changed the title to "Potential impacts of cold frontal passage on air quality over the Yangtze River Delta, China".

7) The figure in the response is interesting, it may be attached to the paper as supplement data.

**Response:** Thanks for your suggestion. This figure is attached as supplement.

Other points should be considered:

Figure 3, red line and green bar, should indicate which is wind speed, which is PM2.5.

**Response:** Red line is wind speed, green bar is $PM_{2.5}$. Please refer to the caption of Figure 3.

Figure 7, profiles of VDIF, ZADV, etc., should indicate over which site or area/zone.

[revised manuscript text omitted]